# Single-cell transcriptome analysis reveals cellular heterogeneity in mouse intra- and extra articular ligaments

Kyota Ishibashi [1,2], Kentaro Ikegami[3], Takashi Shimbo[2,4 ✉], Eiji Sasaki [1], Tomomi Kitayama[2,3], Yuzuru Nakamura [1,2], Takahiro Tsushima[1], Yasuyuki Ishibashi[1] & Katsuto Tamai[2 ✉]

Ligaments are collagenous connective tissues that connect bones. Injury of knee ligaments, namely anterior cruciate ligament (ACL) and medial collateral ligament (MCL), is common in athletes. Both ligaments have important functions, but distinct regeneration capacities. The capacity for recovery after injury also diminishes with age. However, cellular heterogeneity in the ligaments remains unclear. Here, we profiled the transcriptional signatures of ACL and MCL cells in mice using single-cell RNA sequencing. These ligaments comprise three fibroblast types expressing *Col22a1*, *Col12a1*, or *Col14a1*, but have distinct localizations in the tissue. We found substantial heterogeneity in *Col12a1*- and *Col14a1*-positive cells between ACL and MCL. Gene Ontology analysis revealed that angiogenesis- and collagen regulation-related genes were specifically enriched in MCL cells. Furthermore, we identified age-related changes in cell composition and gene expression in the ligaments. This study delineates cellular heterogeneity in ligaments, serving as a foundation for identifying potential therapeutic targets for ligament injuries.

[1] Department of Orthopaedic Surgery, Hirosaki University Graduate School of Medicine, Hirosaki, Aomori, Japan. [2] Department of Stem Cell Therapy Science, Graduate School of Medicine, Osaka University, Suita, Osaka, Japan. [3] StemRIM Inc., Ibaraki, Osaka, Japan. [4] StemRIM Institute of Regeneration-Inducing Medicine, Osaka University, Suita, Osaka, Japan. ✉email: shimbot@sts.med.osaka-u.ac.jp; tamai@gts.med.osaka-u.ac.jp

Ligaments are collagen-rich connective tissues that connect bones to each other. In particular, articular ligaments, as well as tendons, ensure proper bone and muscle deposition at the joints[1]. Functional ligaments are critical for the movement and stability of the joints. Ligament injury compromises efficient movement and results in loss of quality of life[2,3]. The loss of functional ligaments causes various diseases, such as osteoarthritis and intra-articular injury[4–7]. Furthermore, ligament injuries increase with age owing to a decrease in tensile strength, collagen organization, and regeneration capacity of stem cells[8–10]. Thus, maintenance of functional ligaments throughout life is vital for proper locomotion and disease prevention.

The knee is the largest joint in the human body and is stabilized by four major articular ligaments. Specifically, the anterior cruciate ligament (ACL) and posterior cruciate ligament (PCL) are intra-articular ligaments that prevent excessive forward and backward translation of the tibia. As these two intra-articular ligaments do not heal spontaneously, reconstructive surgery is required upon injury[11–13]. In contrast, the extra-articular ligaments, medial collateral ligament (MCL), and lateral collateral ligament (LCL) prevent excessive varus (outward bowing) or valgus (inward) alignment of the knee joints. Extra-articular ligaments have a higher regeneration capacity than intra-articular ligaments[14]. A previous study showed that 32.6% of all sports injuries involve the knee joint[15]. In addition, an epidemiological study of 19,530 sports injuries found that ACL and MCL injuries occurred 20 and 8 times more frequently than PCL or LCL injuries, respectively[16]. Unlike ACL injuries, MCL injuries are usually treated conservatively. However, although both ligaments are known to have different functions and regeneration capacities, the cellular and molecular bases of the differences between the intra- and extra-articular ligaments remain largely unknown.

Recently, single-cell analysis techniques have been recognized as powerful tools to elucidate the physiological functions and maintenance of tissues in health and disease[17–19]. In tendons, single-cell RNA sequencing identified potential tendon stem/progenitor cells (TSPCs) and a regeneration mechanism[20]. However, the cellular composition of ligaments and the basis of the differing regenerative capacities are yet to be elucidated. Thus, in this study, we aimed to provide detailed understanding of the cell composition of each ligament. Here, we used single-cell RNA sequencing to comprehensively profile the cellular composition and heterogeneity of the ACL and MCL in mice. We validated the single-cell RNA-sequencing results using immunostaining and RNA in situ hybridization, and identified distinct fibroblast profiles in these intra-articular ligaments. In addition, we analyzed how aging affects the integrity of the articular ligaments. Our data provide a broad map of knee ligament cell types that may offer new avenues for targeted therapies for injured and aging tissues.

## Results

**Single-cell RNA-sequencing of mouse articular ligaments reveals three distinct fibroblastic populations.** To comprehensively identify the cellular composition of articular ligaments, we analyzed tissues from the ACL and MCL using single-cell RNA sequencing (Fig. 1a). ACL and MCL samples were collected from 10-week-old male C57/BL6J mice and subjected to single-cell RNA-sequencing using a FACS-based method[21]. The data were quality controlled and filtered using Seurat v3[22]. We obtained data on 1,832 ACL and 1,808 MCL cells. Global gene expression profiles for all detected cells were displayed using UMAP (Fig. 1b). Our analysis identified nine clusters, and each cell type was assigned based on the expression of known cell marker genes (Fig. 1c). Clusters 0, 2, and 4 expressing *Pdgfra* and *Col1a1* were

designated as fibroblasts. Endothelial cells expressing *Pecam1* and *Cdh5* (clusters 6 and 8) were detected. Pericytes expressing *Acta2* (cluster 7) were also identified. We also detected cells in the hematopoietic lineage, specifically a macrophage cell cluster (expressing *Adgre1* and *Cd163*) and a neutrophil cell cluster expressing *S100a9*. In general, both ACL and MCL contained the same cell types. However, we observed a substantial difference in the gene expression profiles and fraction of fibroblast clusters derived from the ACL and MCL, suggesting that these two articular ligaments have a distinct cell composition (Fig. 1d, e, Supplementary Data 1).

Next, we investigated the function and role of fibroblasts in the articular ligaments. To achieve this, we extracted fibroblast clusters (clusters 0, 2, and 4) and performed another clustering analysis (Fig. 2a, Supplementary Data 1). After differentially expressed gene (DEG) analysis, we identified three major subclusters in the fibroblast fraction in the ACL and MCL (Fig. 2b). The expression of *Col22a1*, *Col14a1*, or *Col12a1* specifically marked each sub-cluster (Fig. 2c). To understand the functional differences among these sub-clusters, we also compared the expression levels of TSPC and chondrogenic marker genes. Interestingly, TSPC marker genes (*Tppp3, PDGFRα, Lama4, Ly6a*, and *Ly6e*) were mainly expressed in sub-cluster 0 and 2 (*Col14a1*- and *Col22a1*-positive, respectively) (Fig. 2d). In contrast, chondrogenic marker genes (*Sox9, Chad*, and *Acan*) were predominantly expressed in sub-cluster 1 (*Col12a1*+) (Fig. 2e). Next, we examined the localization of each sub-cluster in the ACL and MCL through immunostaining analysis using PDGFRαH2BGFP mice, in which the nuclei of fibroblasts (PDGFRα+) are detectable with histone H2B-fused green fluorescent protein (GFP)[23]. In the MCL, Col22a1 protein, which is a known marker of the junctional zone in tendons (i.e., myotendinous junction)[24], was detected at the tibial attachment (junction of the MCL and tibia) (Fig. 2f). However, Col22a1 protein was evenly distributed in the ACL, unlike in the MCL (Fig. 2f). In the MCL, Col14a1 protein-positive cells were predominantly found outside of the ligaments (Fig. 2g), whereas Col12a1 protein-positive cells were identified both outside and inside (Fig. 2h). We further performed immunostaining with surrogate markers representing *Col12a1*-positive cells (Cd73) or *Col14a1*-positive cells (Cd34), and obtained similar results (Supplementary Fig. 1). We confirmed that most of the Col22a1-, Col14a1-, and Col12a1-positive cells were GFP-positive, indicating that they are fibroblasts. As protein expression in the extracellular matrix (ECM) is not always parallel to RNA expression, we also performed RNA in situ hybridization. Consistent with the immunostaining results, Col14a1 mRNA-positive fibroblasts were detected outside the MCL, whereas Col12a1 mRNA-positive fibroblasts were detected both outside and inside the MCL (Supplementary Fig. 2a, b). To further investigate the functional difference between *Col14a1* (sub-cluster 0)-, *Col12a1* (sub-cluster 1)- and *Col22a1* (sub-cluster 2)-positive cells, we performed gene ontology (GO) analysis (Fig. 2i-k, Supplementary Data 2). Although the pathway "extracellular matrix organization (GO:0030198)" was shared in *Col14a1*- and *Col12a1*-positive cells, "glycosaminoglycan catabolic process (GO:0006027)" and "collagen fibril organization (GO:0030199)" were uniquely identified in *Col12a1*-positive cells (Fig. 2j). In addition, the genes uniquely expressed in *Col14a1*-positive cells were enriched in "cellular response to cytokine stimulus (GO:0071345)," "cytokine-mediated signaling pathway," and "cellular response to growth factor stimulus (GO:0071363)" (Fig. 2i), further suggesting that *Col14a1*- and *Col12a1*-positive cells have distinct functions and roles. Neutrophil-related terms were enriched in *Col22a1*-positive cells unlike in the other cell types (Fig. 2k). These data indicate that articular ligaments consist

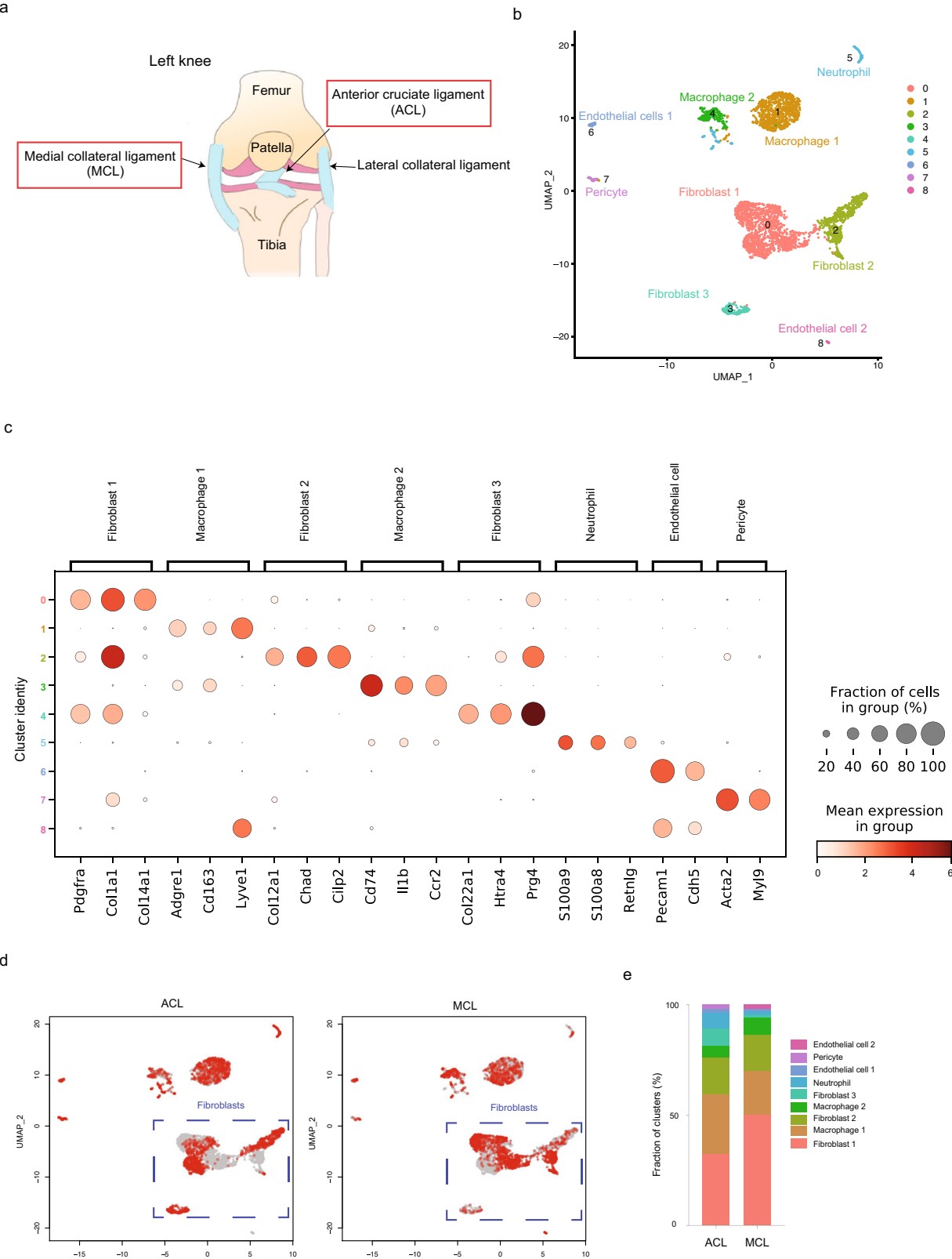

**Fig. 1 Single-cell RNA sequencing of articular ligaments. a** Location of intra- and extra-articular ligaments. **b** Data visualization using a UMAP plot. Nine clusters were identified from 3,640 ligament cells (10 mice). **c** Marker genes for each cluster are shown. Dot size represents the percentage of cells expressing a gene within a cluster. The intensity of the dot color indicates the mean expression level. **d** UMAP plots for ACL and MCL cells. The blue dotted squares indicate fibroblast clusters. **e** Fraction of each cluster in the ACL and MCL.

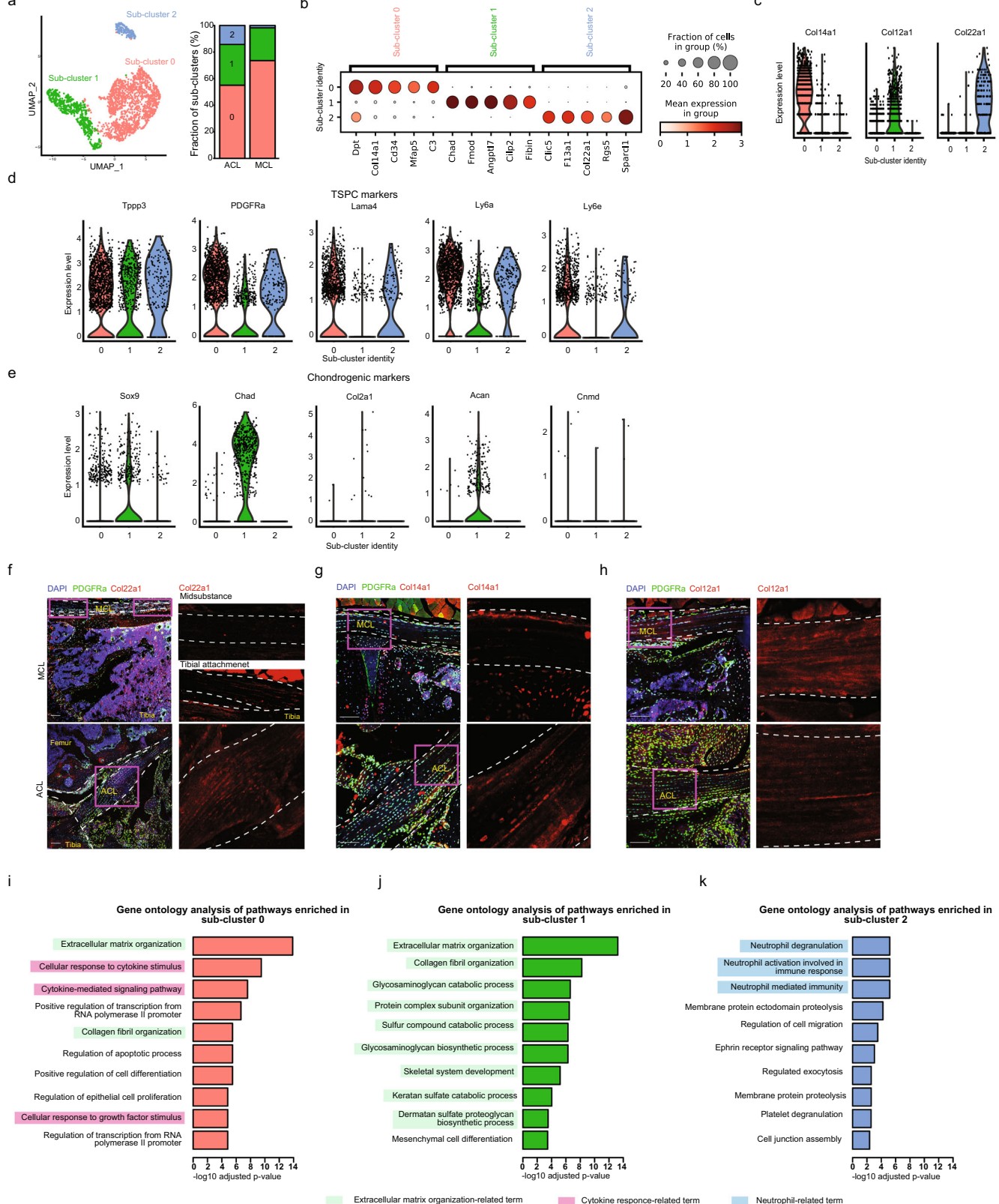

**Fig. 2 Three distinct fibroblastic populations are found in articular ligaments. a** Left, sub-clustering of fibroblasts (cells from clusters 0, 2, and 4 are shown in Fig. 1a). Right, fraction of each sub-cluster in the ACL and MCL. **b** Marker genes for each subcluster are illustrated. Dot size represents the percentage of cells expressing a gene within a cluster. The intensity of the dot color indicates the mean expression level. **c** Violin plots of sub-clusters expressing specific collagens (*Col14a1*, Col12a1, and *Col22a1*). **d, e** Violin plots of tendon stem progenitor cells (TSPC) (**d**) and chondrogenic markers (**e**) in the ACL and MCL. f–h Immunostaining of Col22a1 (**f**), Col14a1 (**g**) and Col12a1 (**h**) in the MCL and ACL. Left, low magnification of the region of interest (ROI) indicated by the magenta square. Right, higher magnification of the ROI. Scale bar: 100 μm. **i–k** Gene ontology analysis shows pathways upregulated in subcluster 0 (*Col14a1*-positive) (**i**), sub-cluster 1 (*Col12a1*-positive) (**j**) and sub-cluster 2 (*Col22a1*-positive) (**k**).

of spatially distinct fibroblasts that are enriched for distinct gene expression patterns associated with specific functions.

**ACL and MCL have a unique fibroblast profile.** DEG analysis between ACL and MCL *Col22*-positive cells identified a set of only 22 genes, suggesting these cells have similar properties. Thus, we performed further detailed analyses focused on *Col14a1*- and *Col12a1*-positive cells to investigate the phenotypic differences between ACL and MCL fibroblasts. UMAP analysis showed transcriptional heterogeneity in both ACL and MCL with respect to both *Col14a1*- and *Col12a1*-positive cell clusters (Fig. 3a). To further delineate the differences, we performed DEG analysis between the ACL and MCL in *Col14a1*- and *Col12a1*-positive cell clusters (Fig. 3b, c). Interestingly, *Tnmd*, a known marker of maturation in tendons, was highly expressed in the *Col12a1*-positive cells in the MCL. Consistent with previous reports[25], we detected protein expression of *Tnmd* in both ACL and MCL (Fig. 3d). However, RNA in situ hybridization analysis showed that *Tnmd* was expressed in the MCL, but not expressed in observable quantities (Fig. 3e). These results indicated that the ACL and MCL consist of fundamentally distinct cell types. To further analyze the ligament-specific features, we performed GO term analysis (Fig. 3f, g). *Col14a1*-positive cells were enriched in terms related to "extracellular matrix organization (GO:0030198)" and "regulation of cell migration (GO:0030334)" in both ACL and MCL. However, only MCL *Col14a1*-positive cells were enriched for "regulation of cell migration involved in sprouting angiogenesis (GO:0090049)," "regulation of angiogenesis (GO:0045765)," and "positive regulation of endothelial cell migration (GO:0010595)" (Fig. 3f). Collagen-related GO terms enriched in *Col12a1*-positive MCL cells included "collagen fibril organization (GO:0030199)," "collagen biosynthetic process (GO:0032964)," and "collagen metabolic process (GO:0032963)". On the contrary, chondroitin-related terms were enriched in the ACL, including "glycosaminoglycan biosynthetic process (GO:0006024)," "chondroitin sulfate biosynthetic process (GO:0030206)," and "chondroitin sulfate metabolic process (GO:0030204)" (Fig. 3g).

***Col12a1*- and *Col14a1*-positive cells display cellular heterogeneity in the ACL and MCL.** Next, we performed further subclustering analysis of *Col12a1*- and *Col14a1*-positive cells to identify unique subpopulations of ACL and MCL fibroblasts. Two sub-populations of *Col14a1*-positive cells, named *Col14a1* type-1 and type-2 fibroblasts, were identified; however, we failed to identify further subpopulations of *Col12a1*-positive cells (Fig. 4a, b, Supplementary Data 1). We also performed subclustering analysis with higher resolution, but did not find additional subpopulations with distinct gene expression signatures (Supplementary Fig. 3a, b). Based on the DEG analysis, *Col14a1* type-1 fibroblasts expressed *Cxcl12*, *C4b*, and *Gas6*. CXCL12 is a well-known chemokine that regulates bone marrow-derived mesenchymal stem cell migration by interacting with *CXCR4*-positive cells[26]. Periodontal ligament studies have also shown that CXCL12 not only regulates cell migration but also promotes angiogenesis in the healing process[27,28]. *Col14a1* type-2 fibroblasts expressed *Dpp4*, *Pi16* and *Procr*. Fibroblasts expressing *Pi16* and *Dpp4* exist across the all tissues and exhibit high expression of stemness-associated genes, *Cd34* and *Ly6a*[29]. These results suggest that the two types of *Col14a1*-positive fibroblasts potentially serve as resource cells and provide a regenerative environment. Furthermore, tenocyte genes (*Tnmd*, *Scx*, *Mkx*, *Tnc*, *Fmod*, and *Thbs4*) were specifically expressed in *Col12a1*-positive cells (Fig. 4d). Interestingly, although the expression levels of *Col1a1* and *Col3a1* were similar between ACL and MCL *Col12a1*-positive cells, tenocyte marker genes were more highly expressed in the

MCL *Col12a1*-positive cells. However, TSPC markers did not show much difference between *Col14a1* type-1 and type-2 cells in the ACL and MCL (Fig. 4e). To further describe the cellular dynamics of these subpopulations, we performed RNA velocity (to predict the future state of a cell) and pseudotime (to order cells along a lineage) analyses (Fig. 4f, g). Interestingly, RNA velocity analysis highlighted a clear distinction between *Col2a1*- and *Col14a1*-positive cells. Pseudotime analysis also showed a substantial change in pseudotime at this border. These data suggest that the two fibroblast populations (*Col2a1*- and *Col14a1*-positive cells) have distinct properties and functions. Furthermore, RNA velocity illustrated a directional flow from *Col14a1* type-1 to *Col14a1* type-2 fibroblasts, suggesting a possibility in which *Col14a1* type-1 fibroblasts differentiate into *Col14a1* type-2 fibroblasts.

**Aging skewed cell composition in the MCL.** We clarified that articular ligaments contain three distinct fibroblast cell types (*Col12a1*-, *Col14a1*- and *Col22a1*-positive cells), and that cell distribution is heterogeneous in the ACL compared to that in the MCL. To determine whether and how fibroblasts in articular ligaments change during aging, we performed additional single-cell RNA sequencing. ACL and MCL samples were isolated from aged (100-week-old) male mice using the same method used for obtaining samples from young mice. Global gene expression profiles for all detected cells were displayed using UMAP (Fig. 5a, Supplementary Data 1). Each cell type was assigned based on the expression of known cell marker genes (Supplementary Fig. 4). In the ACL, the proportion of immune cells (B cells, T cells and neutrophils) increased in the "aged" samples, indicating degeneration of the ACL due to age-related inflammation in osteoarthritis[30,31]. In the MCL, there was little change in the proportion of immune cells between the young and aged samples, whereas the proportion of fibroblasts appeared to differ. We then focused on the *Col14a1*- and *Col12a1*-positive cells in young and aged samples (Fig. 5b). Furthermore, UMAP plots showing the sample origin clearly indicated that aging introduced substantial differences in gene expression in both *Col14a1*- and *Col12a1*-positive cell clusters (Fig. 5c). We found that in the ACL, the proportion of *Col14a1* type-2 cells increased with increasing age, whereas that of *Col14a1* type-1 cells was relatively stable. However, the proportion of *Col14a1* type-1 and type-2 cells decreased with increasing age in the MCL. Overall, the ratio between *Col14a1*- and *Col12a1*-positive cells was remarkably skewed in aged MCL (Fig. 5d, Supplementary Data 1). Immunostaining of Cd34, a marker for *Col14a1*-positive cells, demonstrated that the number of *Col14a1*-positive cells was decreased in aged MCL, whereas the number of *Col12a1*-positive cells (marked by Cd73) was unaffected by aging (Fig. 5e, f, Supplementary Data 1). In addition, we did not observe this skewed pattern in the ratio between *Col14a1*- and *Col12a1*-positive cells in aged ACL (Fig. 5g). To investigate how aging affects gene expression signatures of the cells in the ACL and MCL, we performed DEG analyses between young and aged fibroblasts in the ACL and MCL (Fig. 5h, i). As previously reported[32,33], young ligaments show high expression levels of ECM-encoding genes, such as *Col1a1*, secreted protein acidic and rich in cysteine (*Sparc*), and heat shock proteins (*Hsps*). These aging-induced changes in gene expression were confirmed in human articular ligaments by comparing tissues from young (18-year-old) and aged (82-year-old) individuals. We found that the expression levels of Sparc and Hspa5 were significantly higher in young ligaments than in aged ligaments (Fig. 5j–l, Supplementary Data 1). Overall, we identified both transcriptional and compositional changes during aging in both ACL and MCL.

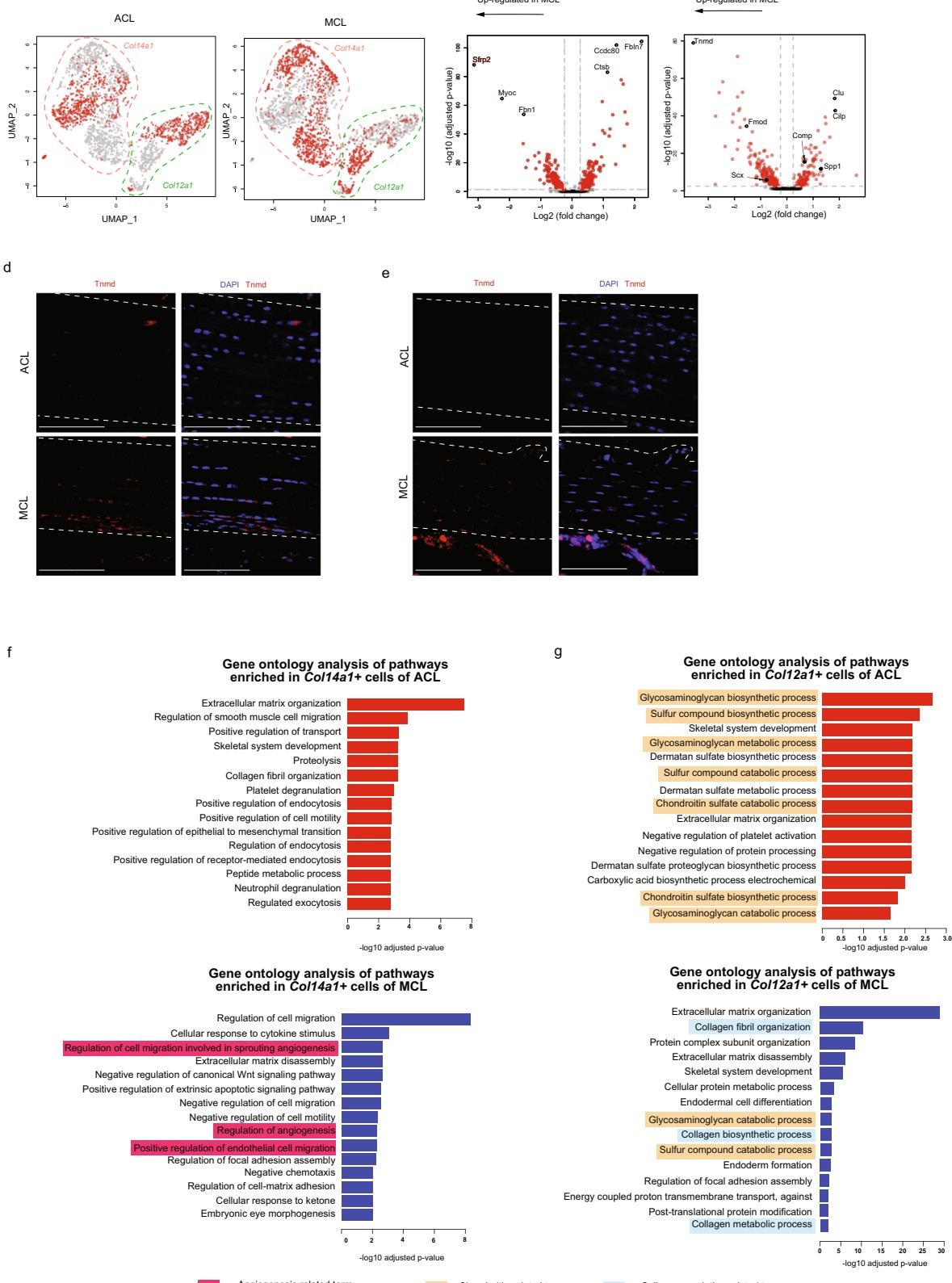

**Fig. 3 Unique fibroblasts in ACL and MCL cells. a** UMAP plots for *Col14a1*- and *Col12a1*-positive cells in the ACL and MCL. **b**, **c** Comparison of differentially expressed genes between the ACL and MCL. Volcano plots for *Col14a1*- (**b**) and *Col12a1*-positive cells (**c**). **d**, **e** Immunostaining (**d**) and RNA in situ hybridization (**e**) of Tnmd in mice ACL and MCL. Scale bar: 100 μm. **f**, **g** Gene ontology analysis comparing *Col14a1*- (**f**) and *Col12a1*-positive cells (**g**) between the ACL and MCL.

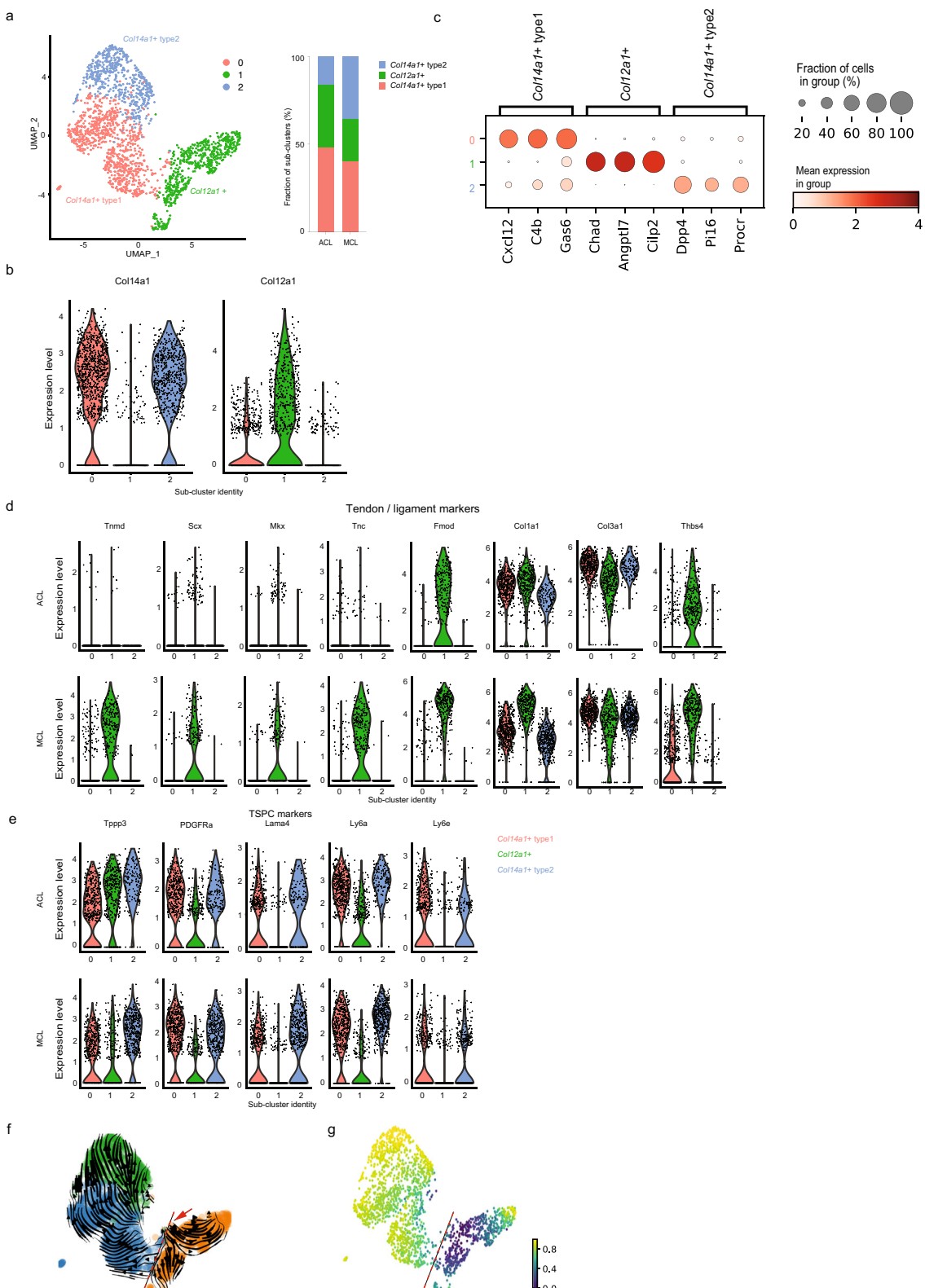

**Fig. 4 Further analysis of *Col12a1*- and *Col14a1*-positive cells reveal cellular heterogeneities in the ACL and MCL. a** Left: further sub-clustering analysis of *Col14a1*- and *Col12a1*-positive cells identified 3 distinct sub-populations. Right: fraction of each sub-cluster in the ACL and MCL. **b** Violin plots of sub-clusters indicate the expression level of *Col14a1* and *Col12a1*. **c** Marker genes for each sub-cluster are illustrated. Dot size represents the percentage of cells expressing a gene within a cluster. The intensity of the dot color indicates the mean expression level. **d** Violin plots of tendon/ligament markers in the ACL and MCL. **e** Violin plots of TSPC markers in the ACL and MCL. **f** RNA velocity analysis showing cellular dynamics in fibroblasts. The red arrow indicates the root of velocity kinetics. The red dot indicates the border by which *Col12a1*-positive cells and *Col14a1*-positive cells are separated. **g** Pseudotime analyses showing inferred ordering of cells.

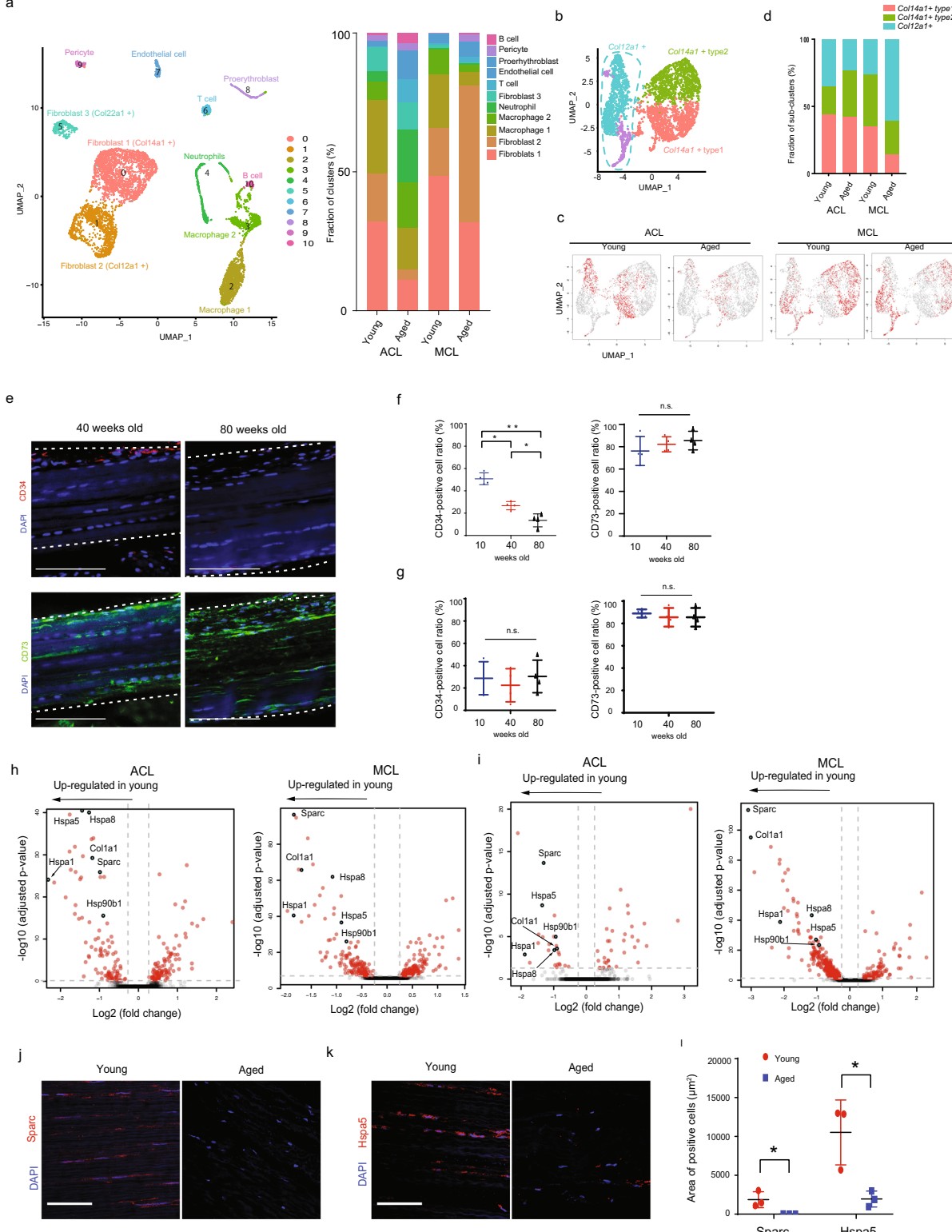

**Fig. 5 Aging impacts gene expression in ligament fibroblasts. a** Left: UMAP plots for articular ligament cells from 10- and 100-week-old mice. (10 mice, respectively) Right: Bar graphs indicate the fraction of each cluster. **b**, **c** UMAP plots for *Col14a1* type-1 and type-2 and *Col12a1*-positive cells from 10- and 100-week-old mice. Merged UMAP plot (**b**). UMAP plots from ACL and MCL, and 10- and 100-week-old mice ligaments (**c**). **d** bar graphs indicate the ratio between *Col14a1* type-1 and type-2 and *Col12a1*-positive cells. **e** Immunostaining of Cd34 and Cd73 in MCLs from 40- and 80-week-old mice. Scale bar: 100 μm. f, g graphs with dot plots indicate the percentage of Cd34- and Cd73-positive cells in the MCL (**f**) and ACL (**g**). *p* values were calculated using one-way ANOVA; *$p < 0.05$. **$p < 0.001$. n = 4. **h**, **i** Comparison of DEGs between young and aged fibroblasts in Col14a1- and Col12a1-positive cells. Volcano plots for *Col14a1*- (**h**) and *Col12a1*-positive cells (**i**). **j**, **k** Immunostaining of Sparc (**j**) and Hspa5 (**k**) in human young (25 years old) and aged (76 years old) ACL ligaments. Scale bar: 100 μm. **l** Expression area of Sparc- and Hspa5-positive cells. p-values were calculated using the Wilcoxon signed-rank test, *$p < 0.05$. $n = 3$.

## Discussion

Ligaments connect bones at various anatomical locations in the body, and many ligament types with unique and distinct properties have been characterized. However, a detailed understanding of the cell composition of each ligament has been previously unavailable. Here, using single-cell RNA sequencing, we identified cellular heterogeneity, in addition to shared properties, in each ligament. In general, the ACL and MCL were composed of the same types of fibroblasts (Col22a1-, Col14a1-, and Col12a1-positive cells). However, we observed ligament-type-specific gene expression patterns upon comparing the ACL and MCL. Furthermore, aging caused massive changes in the cell composition of the MCL as well as decreased expression of several genes encoding ECM proteins in both ligaments.

Using gene expression profiles, we identified three types of fibroblasts, Col22a1-, Col14a1-, and Col12a1-positive cells, in the articular ligaments. We annotated Col22a1-positive cells as junctional fibroblasts, given that such cells are also observed at the myotendinous junction in tendons[24,34]. Although we confirmed Col22a1 expression at the junctional zone (tibial attachment) in the MCL, Col22a1-positive cells were scattered in the ACL. Col22a1-positive cells may also function in the non-junctional area, depending on the ligament type. Alternatively, the ACL may have unique properties in tissue junctions throughout the tissue. The ECM is a major component of the ligament linked to Col14a1- and Col12a1-positive cells. Both Collagen XII and XIV belong to the fibril-associated collagens with interrupted triple helix subfamily, which regulate fibrillogeneisis[35]. In mouse flexor digitorum longus tendons, Collagen XII, which is a mechanosensitive molecule, expresses throughout the tendon with no difference in distribution with age[36]. Col12a1-/- mice displayed discontinuities in 20%–60% of the ACL, suggesting that Col12a1 regulates ligament structure and mechanical properties[37]. Collagen XIV is expressed in areas of high mechanical stress, such as skin, tendon, and cartilage[38,39]. In chicken tendons, the expression level of Collagen XIV was high during embryogenesis, but decreased during tendon maturation[40]. Although protein expression of collagen is not always parallel to RNA expression[40], our study also showed that the proportion of Col14a1-positive cells decreased with age. Although both Col12a1- and Col14a1-positive cells expressed genes encoding ECM synthesis, Col14a1-positive cells showed greater expression of genes related to cytokine or growth factor responses. These distinct gene expression patterns suggest that Col12a1-positive ligament cells, which mainly express Scx, Fmod, and Tnmd, are considered to be "constitutive" cells, the so-called tenocytes in tendons[20]. Furthermore, we identified two types of Col14a1-positive cells (Cxcl12 + and Pi16 + cells). From a fibroblast single-cell atlas composed of 28 datasets[29], Pi16-, Dpp4xi-, Ly6c1xi-, and Col15a1-positive fibroblasts were found in pan-tissues, and pseudotime analysis identified trajectories emerging from Pi16-positive fibroblasts and ending at specialized fibroblasts. These results suggest that Col14a1-positive cells may instead be an "accessory"-type cell, whose role is to help the ligament adapt to the healing environment and regulate constitutive cells. In clinical practice, we should carefully repair the outer zone (containing Col14a1-positive cells) of the MCL, as this region can be critical for the healing capacity of the injured MCL. A limitation of our study was that we examined the expression of only those collagen genes that were used as subtype markers. Further studies need to be performed to elucidate whether the collagen genes described here have specialized functions in each subset.

In general, we found that the cell composition and properties of ligament cells are of tendon cells. This makes sense because both ligaments and tendons are collagen-rich tissues that connect bone-to-bone or bone-to-muscle. In tendons, Tnmd is often referred to as a tissue maturation marker[41,42]. In the neonate mice ACL, Tnmd was expressed at both RNA and protein levels[42,43]. However, we did not detect reliable amounts of mRNA for Tnmd in the matured ACL, although it was certainly expressed in the matured MCL. In addition, although Tnmd is thought to be ubiquitously expressed in tendon and ligament cells, Tnmd was not expressed in all fibroblast clusters, but only in some parts of the MCL clusters. This differential expression of Tnmd between ligaments and tendons suggests that critical differences exist between these tissue types, an avenue for future studies.

The regeneration capacity of the ACL is extremely limited compared to that of the MCL[14]. Several theories regarding the basis of this difference have been reported, including the formation of fibrin-platelet clots, or a lower capacity for proliferation, angiogenesis, and migration in fibroblasts[43–47]. Fibrin-platelet clots play a crucial role as provisional scaffolds at wound sites during the regeneration process. In ACL injury, synovial fluid levels of urokinase plasminogen activators are upregulated, which dissolve fibrin-platelet clots[48]. This extrinsic factor was thought to cause a major difference in regeneration capacity, but none of these theories fully explains why the ACL cannot heal spontaneously. Several animal studies have reported the efficacy of ACL repair using biological enhancements. For example, intra-articular injection of exogenous miR-210 resulted in accelerated ACL healing through upregulation of angiogenesis[49]. In the present study, GO analysis showed that genes uniquely expressed in ACL-dominant cells were enriched for negative regulation of platelet activation (GO:0010544), whereas MCL-dominant cells were enriched for genes that regulate angiogenesis and growth factors. Although we assessed intact ligaments, our results indicate that the ACL and MCL may differ in their intrinsic regeneration capacity, suggesting a requirement for new therapeutic targets for ACL injury. If we improve these factors in the ACL, we can treat ACL injury without reconstruction surgery.

By analyzing articular ligaments from aged mice, we identified further complexities in the cell types found in each ligament. In MCL single-cell RNA-sequencing data, we found that the number of Col14a1-positive cells decreased relative to the number of Col12a1-positive cells. Immunostaining further confirmed that the number of Col14a1 (or Cd34)-positive cells decreased in the tissue, and Cd34 was one of the TSPC markers[20]. In general, aged ligaments and tendons show a decrease in cellularity and disorganization of the ECM, which diminishes the proliferation and differentiation capacities of TSPC[50,51]. In our DEG analysis, Sparc and Col1a1 showed significantly higher expression in young ligaments than in aged ligaments. Sparc-/- mice show decreased expression of ECM-related markers, such as Scx, Mkx, Egr1, and Col1a1[52]. These changes may explain the functional changes in regenerative capacity during aging. However, we did not observe a decrease in the number of Col14a1-positive cells in the ACL, suggesting that each ligament has unique mechanisms to maintain functionality. How these changes in cell composition affect joint function needs to be further examined.

Here, by performing single-cell analysis of the mouse articular ligaments, we were able to provide the first detailed single-cell atlas of articular ligaments. We found that the examined ligaments had unique heterogeneities in terms of cell composition and transcriptional profiles. These profiles showed marked changes in aged mice. Our data serve as a foundation for understanding the properties and functions of the articular ligaments of the knee joint.

## Method

**Mice.** All mice were handled in accordance with the approved guidelines of the Animal Committee of Osaka University Graduate School of Medicine. The protocols were approved by the Animal Committee of the Osaka University Graduate School of Medicine. Mice were housed in cages under a 12 h light/dark cycle. Solid

food and water were provided *ad libitum*. C57BL/6 J mice (10 weeks old) were purchased from CLEA Japan (Tokyo, Japan), and C57BL/6 J mice (100 weeks old) were purchased from the Jackson Laboratory (Tokyo, Japan). PDGFRa-H2B-GFP knock-in mice[23] were purchased from Jackson Laboratory (Bar Harbor, ME, USA).

**Single-cell RNA-sequencing**. For single-cell RNA-sequencing, articular ligaments from male C57BL/6 J mice (10 weeks old) were dissected and subjected to pre-warmed dissociation solution (3 mg/mL collagenase I (Worthington, USA) and 4 mg/mL dispase II (Roche, Tokyo, Japan) in phosphate-buffered saline (PBS)) in a water bath for 1 h at 37 °C. After digestion, the cells were filtered through a 70-μm cell strainer, washed twice in PBS with 2% fetal bovine serum (FBS), and centrifuged at 300× *g* for 5 min to obtain a cell suspension. To obtain live cells, isolated cells were labeled with the following fluorescence-conjugated antibodies through incubation for 20 min at 4 °C: anti-mouse CD45-APC (30-F11; BioLegend, San Diego, USA) and anti-mouse H-2-FITC (AB_1236470; BioLegend). After the antibodies were washed off, SYTOX™ Orange dye (Thermo Fisher Scientific, Waltham, MA, USA) was added for dead cell staining. Live cells (H-2 +, CD45 +, and Sytox − cells) were sorted using the BD FACS Aria III system (BD Biosciences, Franklin Lakes, NJ, USA).

Single-cell RNA sequencing was performed based on a previous report with modifications as follows[53]. Primer mix comprising 5 μL of lysis buffer, 3.1375 μL of Buffer EB (QIAGEN, Hilden, Germany), 0.5 μL of 10 mM dNTP (GenScript, Piscataway, NJ, USA), 0.05 μL of Phusion HF buffer (New England BioLabs, Beverly, MA), 0.3125 μL of Proteinase K (Nacalai Tesque, Kyoto Japana), and 1 μL of 1 μM barcoded oligo-dT primer (5′-ACGACGCTCTTCCGATCT[Barcode] NNNNNNNNNTTTTTTTTTTTTTTTTTTTTTTTTTTTTTTTTTTVN-3′, where "N" is any base and "V" is either "A," "C," or "G"; IDT, Biosystems, Boston, MA, USA) was aliquoted into 384-well plates. The target cells were then sorted into plates. After sorting, the plates were immediately centrifuged and frozen at −80 °C. Plates were incubated at 50 °C for 10 min and then at 80 °C for 10 min. Next, 5 μL of first-strand reaction mix containing 2 μL of 5× SuperScript IV reverse transcriptase (Thermo Fisher Scientific), 0.5 μL of 100 mM DTT (Thermo Fisher Scientific), 0.025 μL of SuperScript IV reverse transcriptase (200 U/μL, Thermo Fisher Scientific), 0.1 μL of SUPERase In RNase Inhibitor (Thermo Fisher Scientific), and 2.375 μL of water were aliquoted into each well. The plates were then incubated at 55 °C for 10 min, and the reaction was inactivated by incubation at 80 °C for 10 min. To remove unincorporated oligos, 2 μL of Exonuclease I mix containing 0.125 μL Exonuclease I (Thermo Fisher Scientific), 1.2 μL 10× Reaction Buffer, and 0.675 μL water was added, and the mixture was then incubated at 37 °C for 20 min. Samples were pooled and purified using DNA Clean & Concentrator Kit-100 (Zymo Research, Orange, CA, USA), concentrated using DNA Clean & Concentrator Kit-5 (Zymo Research), and eluted in 12 μL of Buffer EB. The eluted cDNA was denatured at 95 °C for 2 min and then immediately placed on ice for 2 min. cDNA was pre-amplified using the Accel-NGS 1 S Plus DNA Library Kit (Swift Biosciences, Ann Arbor, MI, USA). Subsequently, the cDNA was incubated at 37 °C for 15 min and at 95 °C for 2 min with 10 μL of Adaptase Reaction Mix containing 1.25 μL of Buffer EB, 2 μL of Buffer G1, 2 μL of Reagent G2, 1.25 μL of Reagent G3, 0.5 of μL Enzyme G4, 0.5 of Enzyme G5, and 0.5 μL Enzyme G6. Next, 23.5 μL of Extension Reaction Mix containing 9.25 μL of Buffer EB, 1 μL of Reagent Y1, 3.5 μL of Reagent W2, 8.75 μL of Reagent W3, and 1 μL of Enzyme W4 was added, and the mixture was incubated at 98 °C for 30 s, at 63 °C for 15 s, and at 68 °C for 5 min. Amplified cDNA was then purified using 26.1 μL of AMPure XP beads (Beckman Coulter Diagnostics, Brea, CA, USA) and eluted in 19.5 μL of Buffer EB. To amplify cDNA libraries, each well was mixed with 3 μL of 10 μM i5 primer (5′-AATGATACGGCGACCACCGAGATCTACAC[i5]ACACTCTTTCC CTACACGACGCTCTTCCGATCT-3′; IDT), 2.5 μL of 12 μM D7 primer (5′-CAA GCAGAAGACGGCATACGAGATCGAGTAATGTGACTGGAGTTCAGACG TGTGCTCTTCCGATC-3′; IDT), and 25 μM KAPA HiFi HotStart ReadyMix (KAPA Biosystems, Boston, MA, USA). Amplification was carried out using the following program: 98 °C for 3 min; 14 cycles of 98 °C for 20 s, 67 °C for 15 s, and 72 °C for 2 min; and a final hold at 72 °C for 5 min. Each well was then purified using 30 μL of AMPure XP beads, eluted in 30 μL of Buffer EB, and transferred to a new PCR tube. Each well was then again purified using 18 μL of AMPure XP beads and eluted in 10 μL of Buffer EB. In total, 1 ng of amplified cDNA was mixed with water to a total volume of 5 μL. Each well was mixed with 10 μL of Nextera TD buffer (Illumina, San Diego, CA, USA) and 5 μL of Amplicon Tagment enzyme (Illumina), and then incubated at 55 °C for 5 min for tagmentation. After tagmentation, the samples were mixed with 20 μL of DNA Binding Buffer (Zymo Research), purified using 32 μL of AMPure XP beads, and eluted in 16 μL of Buffer EB. The eluted DNA was mixed with 2 μL of 10 μM i5 primer, 2 μL of 10 μM P7 primer (5′-CAAGCAGAAGACGGCATACGAGAT[i7]GTCTCGTGGGCTCGG-3′) and 20 μL of NEBNext High-Fidelity 2× PCR Master Mix (New England BioLabs). Amplification was conducted under the following program: 72 °C for 3 min, 98 °C for 30 s, 8 cycles of 98 °C for 10 s, 66 °C for 30 s, and 72 °C for 1 min; and a final hold at 72 °C for 5 min. Sequencing libraries were sequenced on the NextSeq500 or NextSeq 2000 platform. The read length was set to 20 (read 1) + 8 (i7) + 8 (i5) + 51 (read 2) bases.

**Bioinformatics**. Fastq files were mapped to the mouse reference genome (Genome Reference Consortium Mouse Build 38, mm10), and the reads were counted for

each gene using STAR (v2.7.6a)[54]. The output table from STAR was loaded using Seurat v3[55] for further analyses. Low-quality cells were empirically filtered out based on the number of transcripts and genes as well as the percentage of mito-chondrial genes per cell. After filtering, the data were processed using the Seurat standard workflow. DEG analyses were performed using the Seurat "FindMarkers" function. To investigate the functional roles of DEGs, we removed ribosomal protein genes from DEGs and conducted Gene Ontology analyses using the R package Enrichr[56]. The gene expression levels of marker genes were visualized using Scanpy (v1.7.1)[57]. To estimate cellular dynamics, RNA velocity and pseu-dotime analyses were conducted using scVelo[58] and CellRank[59] packages. In the analysis of the scRNA sequencing data of young and old mice, batch effects were corrected using the Seurat "integration" workflow ("SelectIntegrationFeatures," "FindIntegrationAnchors," "IntegrateData") to remove technical artifacts.

**Human intra- and extra-articular ligaments**. ACL samples were harvested from five patients (mean age, 82 years) undergoing lower limb amputation or total knee replacement. For young ligament samples, ACL remnants were harvested from three patients (aged under 20 years) who underwent ACL reconstruction surgery. The samples were fixed in 4% paraformaldehyde (PFA) for 1 day and then placed in OCT compound (Sakura Finetek, Tokyo, Japan) for immunohistochemistry analysis. Ethical approval for this experiment was obtained from the Committee of Medical Ethics of the Hirosaki University School of Medicine Institutional Review Board. Informed consent was obtained from all study participants.

**Immunohistochemistry**. The knee joints from mice were fixed in 4% PFA for 1 day, decalcified in a 0.5 mol/L ethylenediaminetetraacetic acid solution for 10–14 days, and then placed in OCT compound to prepare frozen sections. The samples were sectioned in the sagittal plane at 10 μm and thawed briefly at 20 °C. Permeabilization was performed using 0.5% Triton X-100 (Sigma-Aldrich, Co. LLC., St. Louis, MO, USA), and non-specific antibody binding was blocked with PBS containing 0.1% bovine serum albumin (BSA) and 10% bovine serum (blocking buffer). The samples were incubated with primary antibodies diluted 1:200 in blocking buffer overnight at 4°C and subsequently with fluorescent sec-ondary antibodies diluted 1:500 in blocking buffer overnight for 2 h at 20 °C. The primary antibodies used were rabbit anti-Col22a1 (ab121846; Abcam, Tokyo, Japan), rat anti-Cd34 (ab8158; Abcam), rabbit anti-Cd73 (ab175396; Abcam), rabbit anti-Tnmd (ab203678; Abcam), rabbit anti-Sparc (PA5-78178; Invitrogen, Carlsbad, USA), rabbit anti-Col14a1 (NBP2-15940; Novus Biologicals, Littleton, CO, USA), rabbit anti-Col12a1(BT-AP07935; BT-Lab), and rabbit anti-Hspa5 (PA5-34941; Invitrogen). Antibody staining was visualized using appropriate species-specific secondary antibodies conjugated to Alexa Fluor 647 (A-21244; Invitrogen) and Alexa Fluor 488 (562352; BD Biosciences). Slides were mounted using ProLong Gold anti-fade reagent (Thermo Fisher Scientific) containing the nuclear counterstain 4′,6-diamidino-2-phenylindole. Immunohistochemistry ima-ges were acquired using a Zeiss LSM780 (Germany) confocal microscope and the ZEN Pro 2011 imaging software.

**RNA in situ hybridization**. RNA in situ hybridization was performed using an RNA scope Fluorescent Multiplex Assay kit (ACD Bio-Techne, Newark, CA, USA) and detection probes against Col12a1(312631; ACD Bio-Techne), Col14a1(581941; ACD Bio-Techne), Col22a1(407211; ACD Bio-Techne), and Tnmd (430531; ACD Bio-Techne). The knee joints from mice were fixed in 4% PFA for 1 day, decalcified in a 0.5 mol/L ethylenediaminetetraacetic acid solution for 7 days, and then placed in OCT compound. The samples were sectioned at 6 μm and thawed briefly at 20 °C using Kawamoto films (Section-Lab, Hir-oshima, Japan). Tissue sections were digested with $H_2O_2$ and protease III, hybridized with target probes, amplified, and labeled with fluorophores. Slides were mounted using ProLong Gold anti-fade reagent (Thermo Fisher Scientific) containing the nuclear counterstain 4′,6-diamidino-2-phenylindole and were imaged with the Zeiss LSM780 confocal microscope (Germany) and the ZEN Pro 2011 imaging software.

**Statistics and reproducibility**. In the immunostaining analysis, each group con-tained more than 3 independent samples. Values are expressed as the mean ± standard deviation. Statistical significance was evaluated using the Wilcoxon signed-rank test for comparisons between two groups or ANOVA for multiple comparisons. All statistical analyses were conducted on the JMP Pro software version 13 (SAS Institute Inc., Cary, NC, USA). Statistical significance was set at $p < 0.05$.

**Reporting Summary**. Further information on research design is available in the Nature Research Reporting Summary linked to this article.

## Data availability
All sequencing data used in this study are deposited at the GEO (GSE194427). Source data underlying figures are provided in Supplementary Data 1 (Figs. 1, 2, 4 and 5) and Supplementary Data 2 (Fig. 2).

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

## Acknowledgements

This research was supported by AMED [grant number JP19lm0203018] and JSPS KAKENHI [grant number JP19H03682].

## Author contributions

K.I. (1st author): Conception and design, data collection and assembly, data analysis and interpretation, manuscript writing, and final approval of the manuscript; T.S.: Conception and design, data interpretation, manuscript writing, and final approval of the manuscript; E.S., T.T., and T.K.: Data collection and assembly; K.I. (2nd author): Analysis and interpretation of single-cell RNA sequencing data; Y.I. and K.T.: Conception and design and final approval of the manuscript.

## Competing interests

K.T., K.I. (2nd author), and T.K. disclose the following: K.T. is a scientific founder of and received research funding from StemRIM. K.I., and T.K. are employees of StemRIM. The remaining authors disclose no competing interests.
