## [Peer Review File · Communications Biology]

Reviewers' comments:

Reviewer #1 (Remarks to the Author):

The researchers used single-cell RNA sequencing to analyze the transcriptional signatures of ACL and MCL cells. These ligaments include three fibroblast cell types that express Col22a1, Col12a1, or Col14a1, but have different localizations in the tissue.

This paper lacks depth, and no further mechanism analysis is made. It is recommended to add relevant content.

How is the localization of Col22a1, Col12a1 or Col14a1 determined?

The idea of single-cell sequencing data analysis is more conventional, and it is recommended to analyze pseudo-sequence data. The paper also lacks RNA speed analysis.

Where is the clinical opinion of this paper? Authors are advised to add relevant discussions in the revised manuscript.

Reviewer #2 (Remarks to the Author):

This is a study demonstrating cell heterogeneity in ACL and MCL by using single-cell analysis. Since the information of variety of cell population in tendons and ligaments are missing, this study is very informative. However, the authors need to be carefully interpretate the data based on the previous findings.

Single-cell analysis is very powerful tool to distinguish cell population and RNA sequence data is useful for understanding the current gene expression level. However, in case of ECM genes, these are accumulated in the extracellular milieu after secretion, therefore, the authors need to discuss upon considering that RNA expression may not parallel to the protein level.

The authors mentioned about the tenomodulin expression in discussion, something wrong with the immunostaining. Like I mentioned above, the tenomodulin RNA expression level may decrease, whereas its protein should be expressed in ACL (Sugimoto et al, genesis, 2013).

Regarding to the collagen XII and XIV, these are structurally similar, and their localization has been reported. It has been also reported that collagen XIV is involved in fibrillogenesis, similar to collagen XII, and expression level is decreased with aging (Young et al. Dev. Dyn. 2000, Ansorge et al. H Biol Chem. 2009).

Despite the informative single cell data, it is pity that the discussion does not include the previous information such as molecular interactions, regeneration and repair, mechanical stress, and development and aging. If the authors could include these in discussion, it would demonstrate the significance of this article.

Reviewer #3 (Remarks to the Author):

The authors performed scRNAseq using mouse ACL and MCL samples. They identified three types of fibroblasts, expressing Col14a1, Col12a1, or Col22a1, respectively. Col22a1-positive cells were observed in myotendinous junctions of MCL, and in whole ACL. The authors also examined localization

of Col14a1- and Col12a1-positive cells by IHC. They characterized these subsets by GO analyses. "Glycosaminoglycan catabolic process" and "collagen fibril organization" were uniquely identified in Col12a1-positive cells, while "cellular response to cytokine stimulus," "cytokine-mediated signaling pathway," and "cellular response to growth factor stimulus" were in Col14a1-positive cells. The authors further compared component cells in ACL and MCL and their gene expression profiles. They finally examined ACL and MCL of aged mice, and compared component cells and genes with those of young mice. The approach to examine the differences between ACL and MCL is interesting, and the present findings of fibroblast subsets are interesting; however, this study failed to display specific roles of these subsets clearly.

Major points

1. The authors performed IHC using Col22 antibody, but they used Cd34 and Cd73 antibodies instead of Col14 and Col12. Cd34 has been used as a marker for pericytes, some hematopoietic cells, and some bone marrow cells. Cd73 is a well-known marker for Treg. Expression of Cd34 and Cd73 in these fibroblasts may be specific among fibroblast subsets as shown in Fig. 2b. However, these markers should be expressed in other types of cells. Cd34- or Cd73-positive cells outside the ligaments (Fig. 2f) may not be fibroblasts. At least they should show expression of Cd34 and Cd73 in whole cell population (e.g., Fig. 1c). Why didn't they use Col14 and Col12 antibodies? If there is no available antibodies, the authors should perform co-immunohistochemistry using antibodies for other fibroblast markers, in addition to Cd34 and Cd73.
2. The authors classified the fibroblast population into three subsets, and found respective specific collagen expression. Meanwhile, considering the heterogeneity of these subsets (Fig. 1d, Fig. 3g, h), these subsets should further include subpopulations. In other words, the present classification of three fibroblasts is probably insufficient to reveal specific roles of these subsets. I strongly recommend the authors to further analyze subpopulations in Col14- and Col12-positive fibroblasts, and try to reveal specific subsets which may be involved in specific roles in ACL or MCL. Expression of these collagen is interesting, but I cannot know what the three fibroblast subsets do in ACL and MCL, or in young and aged ligaments after all.
3. Mature ligament cells should be included in the examined cells. They showed Tnmd expression in Fig. 3d-f; however, I could not know which kinds of cells expressed Tnmd. Dot plots or violin plots of other tendon/ligament markers (Tnmd, Scx, Mxk, Tnc, Col3a1, Thbs4, etc) would be useful.
4. Dot plots or violin plots of chondrogenic markers (Sox9, Col2a1, Acan) and various TSPC markers would be also useful for further understanding of three fibroblast subsets.
5. GO analyses were not performed in Col22-positive cells. Considering the specific distribution of Col22-positive cells, the analyses in Fig. 2-4 should be performed using three fibroblast subsets. Or are there any reasons not to examine Col22-positive cells?
6. It is difficult to know the distribution of three fibroblast subsets in the UMAPs of Fig. 3.
7. It would be better to show how distributions of the immune cells and other kinds of cells differ in ACL and MCL, and young and old ligaments. It is possible that ratios of these cells may change depending on the ligaments or aging.
8. Fig. 3g and h display differences of GO in Col14-positive cells and in Col12-positive cells, respectively. Do these data suggest that subpopulations of these fibroblasts are different in ACL and MCL?

Minor points

1. mRNA should be described in italic.

2. How many mice were used for scRNAseq?

Responses to the comments from Reviewer #1

Thank you for your thoughtful and constitutional feedback. We appreciate the editors and reviewers for raising important issues on our manuscript and believe that our revised manuscript has improved considerably by addressing these points. All changes made in the manuscript have been indicated in red font.

1. How is the localization of Col22a1, Col12a1 or Col14a1 determined?

We assessed the localization of *Col22a1*-, *Col12a1*-, and *Col14a1*-positive cells by immunostaining for Col22a1, Cd73 (a surrogate marker for *Col12a1*-positive cells), and Cd34 (a surrogate marker for *Col14a1*-positive cells). In the original manuscript, we used Cd73 and Cd34 antibodies for which we had validated immunostaining protocols. In the revised manuscript, we have added immunostaining analysis using antibodies for Col12a1 and Col14a1 in Fig. 2g, h; and RNA in situ hybridization using probes for Col12a1, Col14a1, and Col22a1 in Supplementary Fig. 2. Thanks to the review, by adding new data, we have now been able to show the correlation between the populations detected by single cell RNA-seq and their locations in each ligament.

2. The idea of single-cell sequencing data analysis is more conventional, and it is recommended to analyze pseudo-sequence data. The paper also lacks RNA speed analysis.

Thank you for your comments. Based on the suggestion, we additionally performed the RNA velocity and pseudotime analyses, and presented the data in Fig. 4e, f. With reference to RNA velocity, we could identify a distinction between *Col12a1*- and *Col14a1*-positive cells, suggesting these two populations have distinct gene expression signatures and, likely, properties. In addition, we now argue the potential differentiation hierarchy within *Col12a1*-positive cells. We believe the additions of new data and discussion has significantly improved the manuscript.

3. Where is the clinical opinion of this paper? Authors are advised to add relevant discussions in the revised manuscript.

Based on your suggestion, we have added the following sentences:

“These results suggest that *Coll4a1*-positive cells may instead be an “accessory”-type cell, whose role is to help the ligament adapt to the healing environment and regulate constitutive cells. In clinical practice, we should carefully repair the outer zone of the MCL, as this may be responsible for the healing capacity of the injured MCL.”

“In the present study, GO analysis showed that genes uniquely expressed in ACL-dominant cells were enriched for negative regulation of platelet activation (GO:0010544), while MCL-dominant cells were enriched for genes that regulate angiogenesis and growth factors. Although we assessed intact ligaments, our results indicate that the ACL and MCL may differ in their intrinsic regeneration capacity, suggesting a requirement for new therapeutic targets for ACL injury. If we improve these factors in the ACL, we can treat injury without reconstruction surgery.”

Responses to the comments from Reviewer #2

Thank you for your thoughtful and valuable feedback. As indicated by the responses that follow, we have taken all of your comments and suggestions into account in the revised version of our manuscript. All changes made in the manuscript are shown in red font.

1. Single-cell analysis is very powerful tool to distinguish cell population and RNA sequence data is useful for understanding the current gene expression level. However, in case of ECM genes, these are accumulated in the extracellular milieu after secretion, therefore, the authors need to discuss upon considering that RNA expression may not parallel to the protein level.

Thank you for your insightful comment. As you suggested, there may be cases in which RNA expression and protein localization are not parallel. We have now described this issue in the discussion. In addition, we performed RNA in situ hybridization using detection probes against Col12a1, Col14a1 and Col22a1, and added the data in Supplementary Fig. 1. We believe that the new in situ RNA data help to map the population detected by single-cell RNA-seq in terms of histology.

2. The authors mentioned about the tenomodulin expression in discussion, something wrong with the immunostaining. Like I mentioned above, the tenomodulin RNA expression level may decrease, whereas its protein should be expressed in ACL (Sugimoto et al, genesis, 2013).

We appreciate the reviewer for raising this important point. Based on the suggestion, we carefully inspected the suggested reference and re-optimized the immunostaining protocol. As the reviewer suggested, this enabled detection of protein expression of Tnmd in the ACL (Fig. 3d). We apologize for the ambiguity of the previous data. In addition, to compare the RNA expression of Tnmd in ACL and MCL, we performed RNA in situ hybridization for Tnmd and found that the expression is very low in the ACL. This RNA in situ hybridization data has been added as Fig. 3e.

These new data encouraged us to carefully perform the detailed analyses of Tnmd expression in human samples. We would like to add further data on human Tnmd properties/functions and report this in a future paper.

3. Regarding to the collagen XII and XIV, these are structurally similar, and their localization has been reported. It has been also reported that collagen XIV is involved in fibrillogenesis, similar to collagen XII, and expression level is decreased with aging (Young et al. Dev. Dyn. 2000, Ansoerge et al. H Biol Chem. 2009).

Thank you for your insightful comment. We assessed the ratio of *Coll4a1*- and *Coll2a1*-positive cells. This study also showed that the proportion of *Coll4a1*-positive cells decreases with age. Based on your comments and previous studies, we have corrected the relevant text in the discussion as follows: “In chicken tendons, the expression level of Collagen XIV was high during embryogenesis, but decreased during tendon maturation³⁶. Although protein expression of collagen is not always parallel to RNA expression, our study showed that the proportion of *Coll4a1*-positive cells decreased with age.”

4. Despite the informative single cell data, it is pity that the discussion does not include the previous information such as molecular interactions, regeneration and repair, mechanical stress, and development and aging. If the authors could include these in discussion, it would demonstrate the significance of this article.

Thank you for your comments. We have improved and added the sentences as follows.

“Both Collagen XII and XIV are interrupted triple helices, which regulate fibrillogenesis³⁰. In mouse flexor digitorum longus tendons, Collagen XII was expressed throughout the tendon with no difference in distribution with age³¹. In contrast, reactivity of Collagen XII in the mature tendon was less than that in postnatal tendons. Collagen XII is a mechanosensitive molecule³², and mechanical stress upregulates Collagen I and matrix assembly³¹. *Coll2a1*^{-/-} mice displayed discontinuities in 20%–60% of the ACL, suggesting that *Coll2a1* regulates ligament structure and mechanical properties³³. Collagen XIV is expressed in areas of high mechanical stress, such as skin, tendon and cartilage^{34 35}. In chicken tendons, the expression level of Collagen XIV was high during embryogenesis, but

decreased during tendon maturation³⁶. Although protein expression of collagen is not always parallel to RNA expression, our study showed that the proportion of *Coll4a1*-positive cells decreased with age. *Coll4a1*-positive cells are required for the expression of transcription factor *Scx*, a marker of tendon progenitor cells in embryonic tendons^{37 38}. In addition, Collagen XIV can bind cell receptors³⁹, suggesting that Collagen XIV regulates other interactions.”

Responses to the comments from Reviewer #3

Thank you for your thoughtful and constitutional feedback. As indicated by the responses that follow, we have taken all of your comments and suggestions into account in the revised version of our manuscript. All changes made in the manuscript are marked in red font.

1. The authors performed IHC using Col22 antibody, but they used Cd34 and Cd73 antibodies instead of Col14 and Col12. Cd34 has been used as a marker for pericytes, some hematopoietic cells, and some bone marrow cells. Cd73 is a well-known marker for Treg. Expression of Cd34 and Cd73 in these fibroblasts may be specific among fibroblast subsets as shown in Fig. 2b. However, these markers should be expressed in other types of cells. Cd34- or Cd73-positive cells outside the ligaments (Fig. 2f) may not be fibroblasts. At least they should show expression of Cd34 and Cd73 in whole cell population (e.g., Fig. 1c). Why didn't they use Col14 and Col12 antibodies? If there is no available antibodies, the authors should perform co-immunohistochemistry using antibodies for other fibroblast markers, in addition to Cd34 and Cd73.

Thank you for your insightful comments. We chose Cd34 and Cd73 antibodies because the distribution patterns of RNA and protein can differ among extra cellular matrix (ECM) genes/proteins. In the revised manuscript, we performed immunostaining of Col14a1 and Col12a1 and confirm that *Col14a1*- and *Col12a1*-positive cells localize differently in the ligaments. The new data is presented in Fig. 2g and h.

Furthermore, based on the reviewer's suggestion, we performed immunostaining of Cd34 and Cd73 using PDGFR α -H2B-GFP knock-in mice. PDGFR α is a well-known marker gene in fibroblasts, detectable by GFP; thus, fibroblasts were labeled with GFP in these mice. New data is added as Supplementary Fig. 2. In addition, we have added the following sentence.

“Next, we examined the localization of each sub-cluster in the ACL and MCL in an immunostaining analysis using PDGFR α H2BGFP mice, in which the nuclei of fibroblasts (PDGFR α +) are detectable with histone H2B-fused green fluorescent protein (GFP).”

Supplementary Fig. 1. Identification of *Col12a1*- and *Col14a1*-positive fibroblasts

a UMAP plots showing the expression of *Col12a1*, *Col14a1*, *Cd34* and *Cd73*, in the ACL and MCL. b,c Immunostaining of Cd73 (b) and Cd34 (c) using PDGFRaH2BGFP mice.

2. The authors classified the fibroblast population into three subsets, and found respective specific collagen expression. Meanwhile, considering the heterogeneity of these subsets (Fig. 1d, Fig. 3g, h), these subsets should further include subpopulations. In other words, the present classification of three fibroblasts is probably insufficient to reveal specific roles of these subsets. I strongly recommend the authors to further analyze subpopulations in *Col14*- and *Col12*-positive fibroblasts, and try to reveal specific subsets which may be involved in specific roles in ACL or MCL. Expression of these collagen is interesting, but I cannot know what the three fibroblast subsets do in ACL and MCL, or in young and aged ligaments after all.

We appreciate for the valuable suggestion. Because we wished to describe major fibroblast subtypes, we focused on three subpopulations in this manuscript. However, we agree with the reviewer in that the fibroblasts could be further divided into smaller subpopulations. To confirm this, we performed further sub-clustering analyses with *Col12a1*- and *Col14a1*-positive cells with several different resolutions (Fig. 4a–d and Supplementary Fig. 3). Under certain conditions, we identified two subtypes of *Col14a1*-positive cells. Although we tested further clustering with higher resolution, we could not detect subpopulations with distinct gene expression signatures (Supplementary Fig. 3). Thus, we concluded that it is not ideal searching for further subpopulations with our data at this point. We would

like to plan new experiments aiming to collect fibroblast with more sequencing depth, and publish the results in another study. In the revised manuscript, we describe identified subpopulations of *Coll4a1*-positive cells and discuss the cell-type specific functions inferred by differentially expressed genes. We also describe potential functional differences between *Coll4a1*- and *Coll2a1*-positive cells.

“Next, we performed further sub-clustering analysis of *Coll2a1*- and *Coll4a1*-positive cells to elucidate unique subpopulations of ACL and MCL fibroblasts. This analysis identified two subpopulations of the *Coll4a1*-positive cells, named *Coll4a1* type-1 and type-2 fibroblasts, while we failed to identify further subpopulation in *Coll2a1*-positive cells, while we failed to identify further subpopulations of *Coll2a1*-positive cells(Fig 4a, b). We also performed sub-clustering analysis with higher resolution, but did not find additional sub-populations with distinct gene expression signatures (Supplementary Fig. 3a, b). Based on the DEG analysis, *Coll4a1* type-1 fibroblasts expressed *Cxcl12*, *C4b* and *Gas6*. CXCL12 is a well-known chemokine that regulates bone marrow-derived mesenchymal stem cell migration by interacting with *CXCR4*-positive cells²⁵. Periodontal ligament studies have also shown that CXCL12 not only regulates cell migration but also promotes angiogenesis in the healing process ^{26 27}. *Coll4a1* type-2 fibroblasts express *Dpp4*, *Pi16* and *Procr*. Fibroblasts expressing *Pi16* and *Dpp4* exist across the all tissues and exhibit high expression of stemness-associated genes, Cd34 and Ly6a²⁸. These results suggest that the two types of *Coll4a1*-positive fibroblasts potentially serve as resource cells and provide a regenerative environment. Furthermore, tenocyte genes (*Tnmd*, *Scx*, *Mkx*, *Tnc*, and *Fmod*) were specifically expressed in *Coll2a1*-positive cells (Fig 4d). Interestingly, although expression levels of *Coll1a1* and *Col3a1* were similar in the ACL and MCL, tenocyte genes were more highly expressed in the MCL.”

Fig 4

Ishibashi K, et al

Supplementary Fig. 3 Further investigation of subpopulations of *Col12a1*- and *Col14a1*- positive cells.

a Left, UMAP plots for sub-clusters of *Col14a1*- and *Col12a1*-positive cells in the ACL and MCL. Right, fraction of each sub-cluster in the ACL and MCL. b Heatmap showing the marker genes for each sub-cluster.

3. Mature ligament cells should be included in the examined cells. They showed *Tnmd* expression in Fig. 3d-f; however, I could not know which kinds of cells expressed *Tnmd*. Dot plots or violin plots of other tendon/ligament markers (*Tnmd*, *Scx*, *Mkx*, *Tnc*, *Col3a1*, *Thbs4*, etc) would be useful.

Based on your suggestion, we have added violin plots of several tendon/ligament markers (*Tnmd*, *Scx*, *Mkx*, *Tnc*, *Fmod*, *Colla1*, *Col3a1*, and *Thbs4*) in Fig4 d. Although *Colla1* and *Col3a1* were expressed

in all sub-populations, tenocyte genes (*Tnmd*, *Scx*, *Mkx*, *Tnc*, *Fmod*, and *Thbs4*) were exclusively expressed in the *Col12a1*-positive cells.

4. Dot plots or violin plots of chondrogenic markers (*Sox9*, *Col2a1*, *Acan*) and various TSPC markers would be also useful for further understanding of three fibroblast subsets.

Based on your suggestion, we have added violin plots of TSPC (*Tppp3*, *PDGFRa*, *Lama4*, *Ly6a*, and *Ly6e*) and chondrogenic (*Sox9*, *Chad*, *Col2a1*, *Acan*, and *Cnmd*) markers in Fig. 2d, e. Interestingly, TSPC marker genes were mainly expressed in sub-cluster 0 and 2 (*Col14a1*- and *Col22a1*- positive, respectively). In contrast, *Sox9*, *Chad*, and *Acan* were predominantly expressed in sub-cluster 1 (*Col12a1*+).

5. GO analyses were not performed in *Col22*-positive cells. Considering the specific distribution of *Col22*-positive cells, the analyses in Fig. 2-4 should be performed using three fibroblast subsets. Or are there any reasons not to examine *Col22*-positive cells?

Thank you very much for pointing out this important issue. We detected only 22 DEGs between ACL and MCL *Col22a1*-positive cells, suggesting that these cells have similar properties in the ACL and MCL. Thus, we focused on *Col12a1*- and *Col14a1*-positive cells. We have added the following sentence:

“DEG analysis between ACL and MCL *Col22a1*-positive cells identified a set of only 22 genes, suggesting these cells have similar properties. Thus, we performed further detailed analyses focused

on *Coll4a1*- and *Coll2a1*-positive cells to investigate the phenotypic differences between ACL and MCL fibroblasts.”

6. It is difficult to know the distribution of three fibroblast subsets in the UMAPs of Fig. 3.

We apologize for the ambiguity in the figure. We have now indicated the distribution of *Coll2a1* and *Coll4a1*-positive cells with dotted circles in Fig. 3a and violin plots of *Coll2a1* and *Coll4a1* in Fig. 4b.

7. It would be better to show how distributions of the immune cells and other kinds of cells differ in ACL and MCL, and young and old ligaments. It is possible that ratios of these cells may change depending on the ligaments or aging.

Based on your suggestion, we have added a UMAP with all cell types and a bar graph indicating the ratio of each cell type in the ACL and MCL, as well as in young and old ligaments, in Fig. 5a. In the ACL, the proportion of immune cells (B cells, T cells, and neutrophils) was increased in the samples from aged mice, indicating degeneration of the ACL due to age-related inflammation in osteoarthritis. In the MCL, there was little change in the proportion of immune cells between the young and aged mouse samples, whereas the proportion of fibroblasts appeared to differ.

Fig 5

Ishibashi K, et al

8. Fig. 3g and h display differences of GO in *Coll14*-positive cells and in *Coll12*-positive cells, respectively. Do these data suggest that subpopulations of these fibroblasts are different in ACL and MCL?

We apologize for the ambiguity in the figure. We have corrected the sentence describing Fig. 3f and g as follows:

“Gene ontology analysis comparing *Coll14a1*- (f) and *Coll12a1*-positive cells (g) between the ACL and MCL.”

Minor points

mRNA should be described in italic.

Based on your suggestion, we have stylized mRNA names in italics.

2. How many mice were used for scRNAseq?

We apologize for the incomplete description in the original manuscript. We collected articular ligament cells for scRNA-seq from 10 mice. To clarify this, we have added the following sentence in the Fig. 1 legend.

“Data visualization using a UMAP plot. Nine clusters were identified from 3,640 ligament cells (10 mice).”

Reviewers' comments:

Reviewer #1 (Remarks to the Author):

I consider this article to be eligible for publication. The authors answered my questions very well.

Reviewer #2 (Remarks to the Author):

This is a revised manuscript submitted by Ishibashi et al. The authors have addressed the issues raised by the first version. The revised manuscript has significant improvements as compared to the original version. However, some information is incorrectly stated and the DISCUSSION is still not clearly written.

Collagens XII and XIV belongs to FACIT subfamily, should not be described as "interrupted triple helices". Fibrillogenesis is spelled incorrectly. Careful attention should be paid to the use of terms and expressions. Although the authors provided the previous information about collagens XII and XIV, the implication of the authors derived from thier data are superficial and lack depth. Therefore, functional difference between the subsets is unclear. If the authors use the collagens as just a subset marker, it would be better to consider it from the characteristics of cell subsets based on the single cell analysis. If the function of the collagens is to be discussed, information of collagen XXII is also needed. Overall, the data is solid, but the discussion is insufficient, and the significance of this study is unclear.

Reviewer #3 (Remarks to the Author):

The authors generally addressed the issues.

1. Although I did not point out in the first review, it would be better to add descriptions in the Title, the Abstract, and the last part of Introduction to make it clear that mouse samples were used. For example, "Single-cell transcriptome analysis reveals cellular heterogeneity in intra- and extra-articular ligaments of the mouse knee joint".
2. I understand that gene expression profiles of Col22a1-positive cells in ACL and MCL are similar. However, GO analysis of Col22a1-positive cells vs other two cells would be useful to illustrate the characteristics of Col22a1-positive cells, as the authors showed Fig. 2i&j for Col14a1- and Col12a1-positive cells.
3. It is interesting that Col14a1-type-2 cells are more abundant in MCL, while the number of Col14a1-type-1 cells are similar in MCL and ACL (Fig. 4a). To better know their features, TSPC markers shown in Fig. 2d should be also compared in Fig. 4.
4. How about the ratio of Col14a1-type-1 and type-2 cells in ACL and MCL of young and aged mice? These data may enrich the significance of the present findings.

Reviewers' comments:

Reviewer #1 (Remarks to the Author):

I consider this article to be eligible for publication. The authors answered my questions very well.

We are thankful to Reviewer #1 for considering our revised manuscript suitable for publication.

Reviewer #2 (Remarks to the Author):

Collagens XII and XIV belongs to FACIT subfamily, should not be described as “interrupted triple helices”.

We apologize for the inaccurate description. We have corrected it in the revised manuscript.

Fibrillogenesis is spelled incorrectly. Careful attention should be paid to the use of terms and expressions.

We apologize for this typo. We have corrected the spelling, and additionally, checked the entire manuscript for any such typos or misspellings.

Although the authors provided the previous information about collagens XII and XIV, the implication of the authors derived from their data are superficial and lack depth. Therefore, functional difference between the subsets is unclear. If the authors use the collagens as just a subset marker, it would be better to consider it from the characteristics of cell subsets based on the single cell analysis. If the function of the collagens is to be discussed, information of collagen XXII is also needed.

We appreciate the critical point given by the reviewer. We have used the collagen genes *Col12a1*, *Col14a1*, and *Col22a1* as subset markers. We need to perform further experiments to elucidate the function of each collagen in each subset. We would like to conduct those experiments in future studies and present the findings in a separate article. We have now clearly stated this in the revised manuscript.

In our previous revised manuscript, based on the reviewer's comment, we have added content related to collagen genes in the discussion section. Thanks to the reviewer's suggestion, we believe that the manuscript has improved. However, as the reviewer pointed out in the second review, there were too many speculations as we did not have data to discuss the functional differences. Thus, we have re-organized and minimized the discussion in the current manuscript. In addition, in the revised

manuscript, we have mentioned the need to perform further experiments to clarify the functional differences.

Reviewer #3 (Remarks to the Author):

1. Although I did not point out in the first review, it would be better to add descriptions in the Title, the Abstract, and the last part of Introduction to make it clear that mouse samples were used. For example, “Single-cell transcriptome analysis reveals cellular heterogeneity in intra- and extra-articular ligaments of the mouse knee joint”.

According to this suggestion, we have now modified our revised manuscript (text in red in title, abstract, and end of introduction).

2. I understand that gene expression profiles of *Col22a1*-positive cells in ACL and MCL are similar. However, GO analysis of *Col22a1*-positive cells vs other two cells would be useful to illustrate the characteristics of *Col22a1*-positive cells, as the authors showed Fig. 2i&j for *Col14a1*- and *Col12a1*-positive cells.

We thank the reviewer for pointing this out. We have re-calculated and identified uniquely expressed genes in *Col22a1*-positive cells. Using the list of genes, we have also performed GO analysis and included the data in Figure 2k. Interestingly, GO terms related to neutrophils were enriched in *Col22a1*-positive cell specific genes, unlike in the other cell types. Consistent with the findings of previous publications, this GO analysis suggests that *Col22a1*-positive cells have unique and distinct functions and properties in fibroblasts in ligaments. We have supplied the list of differentially expressed genes among the cell types and the results of GO analysis as Supplementary data 1.

3. It is interesting that *Col14a1*-type-2 cells are more abundant in MCL, while the number of *Col14a1*-type-1 cells are similar in MCL and ACL (Fig. 4a). To better know their features, TSPC markers shown in Fig. 2d should be also compared in Fig. 4.

We have added a new figure, Figure 4e, showing the expressions of TSPC markers (*Tppp3*, *Pdgfra*, *Lama4*, *Ly6a*, and *Ly6e*). We did not observe much difference in the expression of TSPC markers between *Col14a1*-type-1 and type-2 cells. Given that these cell types are subclusters derived from one cluster, expressing similar cell type markers is reasonable. We speculate that these cells have functionally distinct roles, but we would like to investigate their detailed functions in future studies.

4. How about the ratio of *Col14a1*-type-1 and type-2 cells in ACL and MCL of young and aged mice? These data may enrich the significance of the present findings.

Thank you for this interesting question. We have re-analyzed our data to approach this question. We found that the proportion of *Col14a1*-type-2 cells increased with aging, whereas the proportion of *Col14a1*-type-1 cells was relatively stable in the ACL. However, with increasing age, the proportion of *Col14a1*-type-1 and *Col14a1*-type-2 cells decreased in the MCL. These data suggest that the subtypes function differently in different ligaments or during different developmental time periods. We have added this data as Figure 5b and 5d and the description in the Results section.

REVIEWERS' COMMENTS:

Reviewer #3 (Remarks to the Author):

The authors appropriately revised the manuscript.